# Pedestrian Motion Reconstruction: A Large-scale Benchmark via Mixed Reality Rendering with Multiple Perspectives and Modalities

**Yichen Wang[1], Yiyi Zhang[1]\*, Xinhao Hu[1], Li Niu[1], Jianfu Zhang[1]\*, Yasushi Makihara[2], Yasushi Yagi[3], Pai Peng[4], Wenlong Liao[4], Tao He[4], Junchi Yan[1], Liqing Zhang[1]\***

[1]Department of CSE & MoE Key Lab of AI, Shanghai Jiao Tong University
[2] Independent Researcher, Osaka, Japan
[3] Department of Intelligent Media, SANKEN, Osaka University
[4] Cowa Tech. Ltd.
`{rachal, yi95yi, c.sis, yanjunchi}@sjtu.edu.cn, zhang-lq@cs.sjtu.edu.cn`
`https://github.com/coding-rachal/PMRDataset`

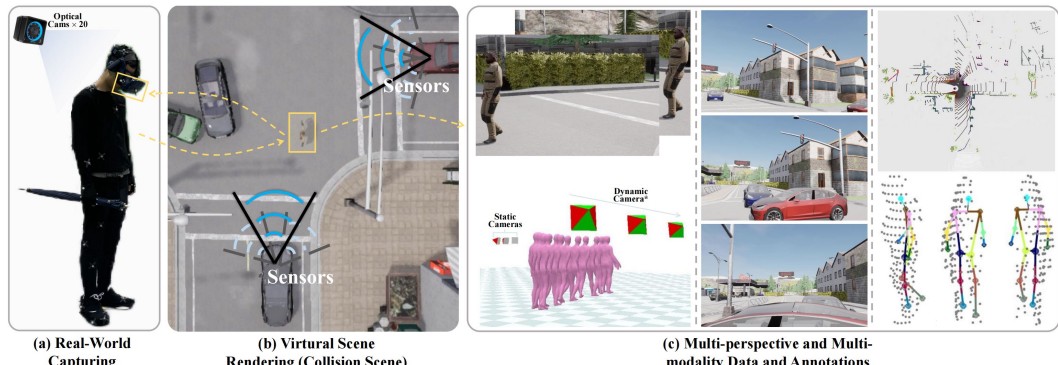

(a) Real-World Capturing  (b) Virtural Scene Rendering (Collision Scene)  (c) Multi-perspective and Multi-modality Data and Annotations

Figure 1: (a-b) **Pedestrian Motion Reconstruction** via mixed reality rendering system. (c) Multi-perspective and multi-modality data collection (Left: Two third-person perspectives with pedestrian motion in global coordinates. Mid: Egocentric perspective of pedestrian. Right: LiDAR modality and skeleton annotation from (a).)

## Abstract

Reconstructing pedestrian motion from dynamic sensors, with a focus on pedestrian intention, is crucial for advancing autonomous driving safety. However, this task is challenging due to data limitations arising from technical complexities, safety, and cost concerns. We introduce the Pedestrian Motion Reconstruction (PMR) dataset, which focuses on pedestrian intention to reconstruct behavior using multiple perspectives and modalities. PMR is developed from a mixed reality platform that combines real-world realism with the extensive, accurate labels of simulations, thereby reducing costs and risks. It captures the intricate dynamics of pedestrian interactions with objects and vehicles, using different modalities for a comprehensive understanding of human-vehicle interaction. Analyses show that PMR can naturally exhibit pedestrian intent and simulate extreme cases. PMR features a vast collection of data from 54 subjects interacting across 12 urban settings with 7 objects, encompassing 12,138 sequences with diverse weather conditions and vehicle speeds. This data provides a rich foundation for modeling pedestrian intent through multi-view and multi-modal insights. We also conduct comprehensive benchmark assessments across different modalities to thoroughly evaluate pedestrian motion reconstruction methods.

---

\* Corresponding author.

Table 1: Comparative statistics of Human Motion datasets.

| Type | Dataset | LiDAR | Ego-Video | 3rd-Video | | Human Pose | | #Frames* | #Subjects | #Scenes | Rare Scenes |
| --- | --- | --- | --- | --- | --- | --- | --- | --- | --- | --- | --- |
| | | | | Camera Modes | Camera Quantity | 3d kpts | mesh | | | | |
| Universal Human Motion | Human3.6M (Ionescu et al., 2014) | ✗ | ✗ | Static | Multi | ✓ | ✓ | 900*/3600k** | 11 | 7 | ✗ |
| | 3DPW (von Marcard et al., 2018) | ✗ | ✗ | Dynamic | Single | ✓ | ✓ | 51k | 7 | - | ✗ |
| | PoseTrack (Andriluka et al., 2018) | ✗ | ✗ | Static | Single | ✗ | | 23k | - (Wild) | - (Wild) | ✗ |
| | GTA-IM (Cao et al., 2020) | ✗ | ✗ | Static | Multi | ✓ | ✗ | 1000k** | 50 | 10† | ✗ |
| | EgoBody (Zhang et al., 2021) | ✗ | ✓ | Both‡ | Multi | ✓ | ✓ | 220k | 36 | 10 | ✗ |
| | Kinpoly-M (Luo et al., 2021) | ✗ | ✓ | - | - | ✓ | ✓ | 148k | - | 5† | ✗ |
| | GIMO (Zheng et al., 2022) | ✗ | ✓ | - | - | ✓ | ✓ | 129k | 11 | 19 | ✗ |
| | RICH (Huang et al., 2022) | ✗ | ✗ | Both‡ | Multi | ✓ | ✓ | 540k | 22 | 5 | ✗ |
| | SLOPER4D (Dai et al., 2023) | ✓ | ✓ | Dynamic‡ | Single | ✓ | ✓ | 100k | 12 | 10 | ✗ |
| | BEDLAM (Black et al., 2023) | ✗ | ✗ | Static | Single | ✓ | ✓ | 380k | - | 8† | ✗ |
| | RELI11D (Yan et al., 2024) | ✓ | ✗ | Static | Single | ✓ | ✓ | 239k | 10 | 7 | ✗ |
| | HiSC4D (Dai et al., 2024) | ✓ | ✗ | - | - | ✓ | ✓ | 36k | 8 | 4 | ✗ |
| | EHPT-XC (Cho et al., 2024) | ✗ | ✗ | Static | Single | ✓ | ✗ | 16k | 82 | 158 | ✓ |
| | MMVP (He et al., 2024) | ✗ | ✗ | Static | Single | ✓ | ✓ | - | - | 10 | ✗ |
| | HmPEAR (Lin et al., 2024) | ✓ | ✗ | Static | Multi | ✓ | ✓ | 250k | 25 | 10 | ✗ |
| Pedestrian Motion | PIE (Rasouli et al., 2019) | ✗ | ✗ | Dynamic | Single | ✗ | ✗ | 293k | -(Wild) | -(Wild) | ✗ |
| | Euro-PVI Bhattacharyya et al. (2021) | ✓ | ✗ | Dynamic | Single | ✗ | ✗ | 83k | -(Wild) | -(Wild) | ✗ |
| | nuScenes (Caesar et al., 2020) | ✓ | ✗ | Dynamic | Multi | ✗ | ✗ | 40k | - (Wild) | -(Wild) | ✗ |
| | Waymo Open (Sun et al., 2020) | ✓ | ✗ | Dynamic | Multi | ✓ | ✗ | 200k | - (Wild) | -(Wild) | ✗ |
| Both | PMR (ours) | ✓ | ✓ | Both | Multi | ✓ | ✓ | 225k*/1355k** | 54 | 12† | ✓ |

Note: Kinpoly-M: Kinpoly-Mocap, *: computed from unique seqs, **: computed from multiview third-person perspective seqs, †: virtual scenes, ‡:the parameters of dynamic cameras are unknown, ⋆: we only count for annotated frames.

# 1 INTRODUCTION

Pedestrian motion reconstruction plays a crucial role in deciphering the dynamics of interactions between pedestrians and vehicles, serving as a foundational element for numerous applications, including autonomous driving (Prédhumeau et al., 2021; Camara et al., 2020a;b), assistive technologies (Zhou & Hu, 2008), *etc*. This task requires effective modeling of pedestrian intention. The intention behind human motion indicates the future movement of pedestrians, which is essential for understanding pedestrian behavior. Human intention is complex and comprises multiple factors, including environmental conditions and the pedestrian's perception/judgment of the environment. Pedestrian motion estimators in autonomous driving systems require abundant data to make robust decisions and comprehensive sensor observations to avoid ignoring critical environmental factors. Hence, data collection is crucial in this area for ensuring safe autonomous driving systems.

However, data collection for pedestrian motion reconstruction faces two key challenges: **Difficulty for High-quality Annotation:** Achieving precise depictions of pedestrian motions within a uniform global coordinate system is particularly challenging in real-world scenarios, leading to the limited scale of current datasets (Saini et al., 2019). **Rare Scenarios and Safety Concerns:** Another significant challenge is the safety and financial implications of documenting human behavior in complex scenarios involving vehicles, such as collisions or pedestrian accidents. Conducting real-world data collection in these rare events requires substantial investment in setting up the necessary environments, acquiring equipment, and employing actors, aside from the direct risks to participants involved (Kim et al., 2019; Ettinger et al., 2021; Sun et al., 2019; Cong et al., 2022).

To address the previously highlighted challenges, we present the Pedestrian Motion Reconstruction (PMR) dataset, a comprehensive resource designed for intention-aware pedestrian motion reconstruction using data from moving sensors. PMR integrates real-world behaviors with high-quality, labor-free annotations from simulations to capture complex interactions between pedestrians, objects, and vehicles, especially in rare or safety concern scenarios. Collected using a mixed-reality platform combining a VR headset, MoCap system, and CARLA simulator (Dosovitskiy et al., 2017), PMR reduces data collection costs and risks while generating realistic sequences from moving sensors like cameras and LiDAR, ensuring ground-truth alignment with the global coordinate system and capturing the pedestrian's egocentric perspective. Additionally, as human intention is complex and comprises multiple factors, including environmental conditions and the pedestrian's perception and judgment of the environment, this multi-modal and multi-view approach comprehensively and systematically enriches the representation of pedestrian intent under complex environments. Based on our analyses, PMR showcases pedestrians expressing reasonable reactions, particularly in dangerous scenarios (*i.e.*, when they are close to vehicles). *To the best of our knowledge, PMR is the first large-scale human motion dataset to incorporate multi-perspectives and multi-views to model pedestrian intention in diverse outdoor scenes, including rare scenarios such as collisions with safety concerns.*

This allows for comprehensive analysis and understanding of pedestrian intent, the ability to simulate extreme scenarios, and effective benchmarking.

**The contributions of this benchmark are as follows:** (1) We present PMR, a large-scale, comprehensive dataset derived from an innovative mixed reality platform that includes third-person perspective RGB videos from moving vehicles, LiDAR data from vehicles, and the egocentric perspective of pedestrians. This large-scale dataset directly addresses current technical difficulties, modeling challenges, and concerns related to cost and safety, effectively bridging the gap in pedestrian motion reconstruction research. (2) We evaluate the dataset across multiple perspectives and modalities: (a) **Global Pedestrian Pose Reconstruction from Moving Cameras**: reconstructing pedestrian meshes in a unified global coordinate system from monocular or multi-view RGB videos captured from a third-person perspective by one or more moving vehicles; (b) **Head Pose Estimation from Egocentric Perspective Video**: determining the head pose of pedestrians from their viewpoint recorded via a VR headset; (c) **Multimodal Pedestrian Pose Estimation**: estimating 3D pedestrian keypoints from third-person perspective RGB videos taken by moving cameras and corresponding LiDAR data. Our experiments demonstrate the significant impact of our dataset on analyzing pedestrian intent. (3) We evaluate the real synthesis domain gap across multiple downstream tasks.

## 2    RELATED WORKS

### 2.1    PEDESTRIAN MOTION ESTIMATION

Pedestrian motion estimation is crucial for autonomous driving. 3D pose estimation is an effective approach to estimate pedestrian motion. Most existing methods rely on calibrated, synchronized, and static multi-view capture setups. Jointly estimating human pose and camera motion from videos captured by moving cameras is challenging due to the entangled human and camera motions. Huang *et al*. (Huang et al., 2021) use uncalibrated cameras but assume temporal synchronization and static configurations. Hasler *et al*. (Hasler et al., 2009) handle unsynchronized moving cameras using multi-view input and audio stream synchronization. Dong *et al*. (Dong et al., 2020) reconstructs 3D poses from unaligned internet videos of various actors, assuming multiple viewpoints of the same pose. Luvizon *et al*. (Luvizon et al., 2023) estimate global human poses using scene point clouds with static cameras. Few methods attempt to estimate human pose in global coordinates from monocular videos recorded with dynamic cameras (Li et al., 2022; Yuan et al., 2021a; Ye et al., 2023b). However, they do not consider the egocentric perspective, which reflects human intention, as it is impractical to collect in real-world scenarios.

### 2.2    3D HUMAN MOTION DATASETS

Many datasets have been proposed to facilitate research on 3D human pose estimation. As shown in Table 1, we categorize 3D human motion datasets into universal human motion datasets and pedestrian motion datasets. H36M (Ionescu et al., 2014) provides synchronized video with optical-based MoCap in studio environments. 3DPW (von Marcard et al., 2018) provides 3D annotations in the wild using a single hand-held RGB camera. MuPoTS-3D (Mehta et al., 2017) focuses on multi-person scenes with occlusions from multi-view capture. However, none of these datasets provide global motion annotations of pedestrians. Egobody (Zhang et al., 2021) provides the global motion of the interactee from a head-mounted camera, but not the interactor. RICH (Huang et al., 2022) uses seven static cameras and one moving camera, but lacks ground truth for camera pose. SLOPER4D (Dai et al., 2023) is the first scene-natural 3D human motion dataset captured with wearable sensors, but it struggles with capturing complex and extreme scenarios, such as collisions.

Pedestrian action and intention are also important in traffic scenes. However, existing autonomous driving datasets (Bhattacharyya et al., 2021; Caesar et al., 2020; Rasouli et al., 2019; Black et al., 2023; Sun et al., 2020; Chang et al., 2019; Houston et al., 2021; Chandra et al., 2019; Malla et al., 2020), have limited annotations and analysis on pedestrian-vehicle interactions. Datasets like nuScenes (Caesar et al., 2020), Argoverse (Chang et al., 2019), and Lyft L5 (Houston et al., 2021) focus on vehicle trajectories with sparse pedestrian interactions. PIE (Rasouli et al., 2019), Euro-PVI (Bhattacharyya et al., 2021) include diverse vehicle-pedestrian interactions but only provide 2D or 3D bounding boxes. Waymo (Sun et al., 2020) added 3D keypoint annotations, but they are estimated from noisy point clouds. Such capturing methods prevent simultaneous pedestrian-view

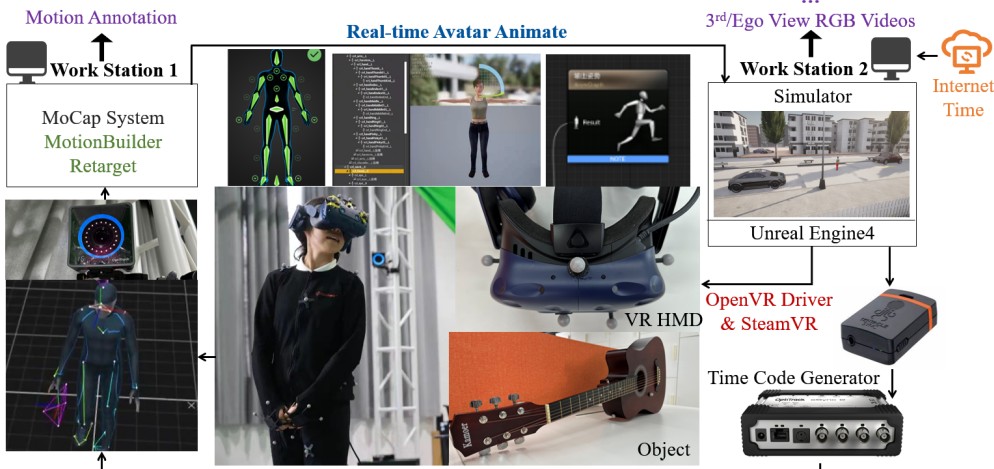

Figure 2: Our pedestrian motion collection system pipeline based on VR and MoCap. The real-time position and pose of characters and objects (middle) are obtained by the MoCap system (left) and provided to the counterparts in the simulated environment (right top). Time synchronization (right bottom) is taken between devices.

RGB videos, accurate 3D keypoints, and third-person views, and avoid high-risk, high-cost scenarios. In contrast, simulators built on graphical engines provide realistic simulations for autonomous driving (Dosovitskiy et al., 2017; Shah et al., 2017), offering third-person perspectives and egocentric views with accurate global pose annotations without real-world risks. This approach leverages realistic sensor simulation to overcome traditional data collection challenges. Our PMR dataset uses this technology to provide multi-modal and multi-perspective data with accurate and comprehensive motion annotations in carefully designed scenes, including rare scenarios. This enables universal human motion reconstruction and pedestrian-specific analyses.

## 3 PMR DATASET CONSTRUCTION

### 3.1 MIXED REAL/VIRTUAL RENDERING PLATFORM

We have developed a mixed real/virtual rendering platform along with a pedestrian motion collection system to facilitate our data acquisition process. The complete collection pipeline is illustrated in Fig. 2. In our setup, volunteers are represented by virtual avatars in the Unreal Engine 4 (UE4) environment, experiencing our designed scenes through VR Head-Mounted Displays (HMDs). This setup induces a series of reactions and interactions from the volunteers. Their movements are captured using an optical motion capture system and processed through a dedicated pipeline to drive avatar actors in real-time within the UE4 engine (Fig. 3).

To further enhance the realism of human-object interactions with high quality labor-free labels, we incorporate real-world objects (e.g., suitcase, handbag) into virtual environments through motion capture system (Fig. 4), enabling volunteers to engage with these objects and enrich the variety and naturalness of their movements.

We design 12 distinct scenarios with varying vehicle approach speeds, including rare but critical cases to study pedestrian behavior during interactions with vehicles (e.g., emerging from blind spots, witnessing collisions) 5. These scenarios include RGB videos from vehicle sensors, egocentric perspective RGB videos, vehicle sensor parameters, and LiDAR data. Moreover, we extract SMPL-X (Loper et al., 2015; Pavlakos et al., 2019) representations of the volunteers' motions from the raw optical motion capture recordings using Mosh++ (Mahmood et al., 2019; Loper et al., 2014). For further details regarding the hardware, configurations, and data collection pipeline, please refer to the *supplementary materials*.

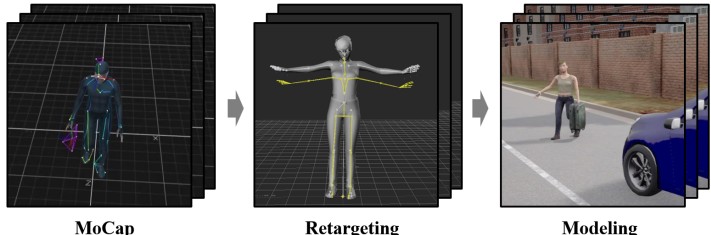

Figure 3: Pipeline of Avatar MoCap, Motion Retargeting, and Modeling. In the retargeting step, it's recommended to ensure the best possible alignment of the two skeletons(the yellow one and the gray one). The situation here is for illustration only.

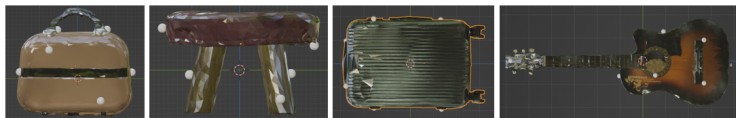

Figure 4: Simulating marks and adjusting models' centers to facilitate reflecting transformations of real objects in the virtual environment accurately.

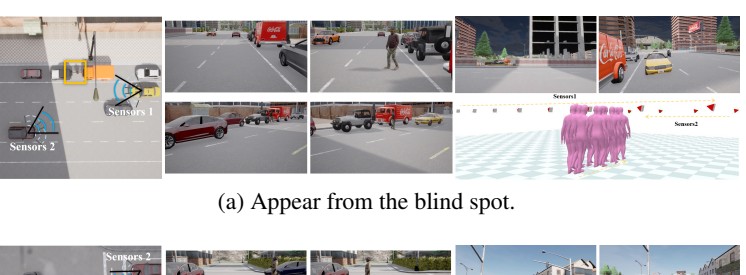

(a) Appear from the blind spot.

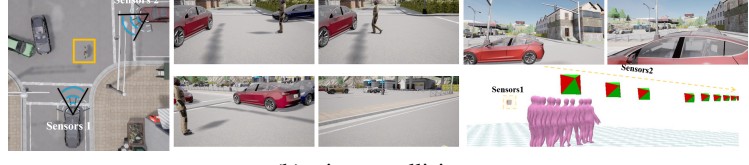

(b) witness collision.

| Bird eye's view of each scene | 3rd-view image sequence from Sensors 1 | Ego-view image sequence |
| | 3rd-view image sequence from Sensors 2 | Annotations |

Figure 5: Example scenarios in the dataset.

Table 2: Statistics of our PMR Dataset. #Sequences: Multiview 3rd-perspective sequences.

| #Sequences | #Frames | #IDs | #Scenarios | #Objects | Average FPS |
|---|---|---|---|---|---|
| 12,138 | 1,355,064 | 54 | 12 | 7 | 6.72 |

## 3.2 STATISTICS AND ANALYSIS OF PMR

Leveraging our platform, we collected an extensive dataset of labeled pedestrian motion data, designated as the PMR dataset. A total of 54 volunteers participated in our data collection efforts. Each volunteer selected several pre-designed scenes and interacted with the environment while equipped with a motion capture suit and a VR HMD. The data acquisition protocol involved recording 9 sequences for each chosen scene, incorporating 3 distinct weather conditions and 3 specific car modes within each scene. Each recorded sequence contains: 6 third-view RGB videos (with different perspectives and camera speeds), 1 egocentric video, and 2 LiDAR point sequences, as well as real-time annotation-free accurate labels (*i.e.*, (SMPL-X (Loper et al., 2015), 3D/2D skeletons and bounding boxes, third-view camera extrinsic, semantic lidar labels, and lidar parameters) from MoCap and CARLA simulator, as summarized in Fig. 1. The PMR dataset is notable for its diversity, encompassing various characters, weather conditions, and scenes. This diversity renders it a valuable resource for investigating a wide array of scenarios and applications. Detailed statistics, including the

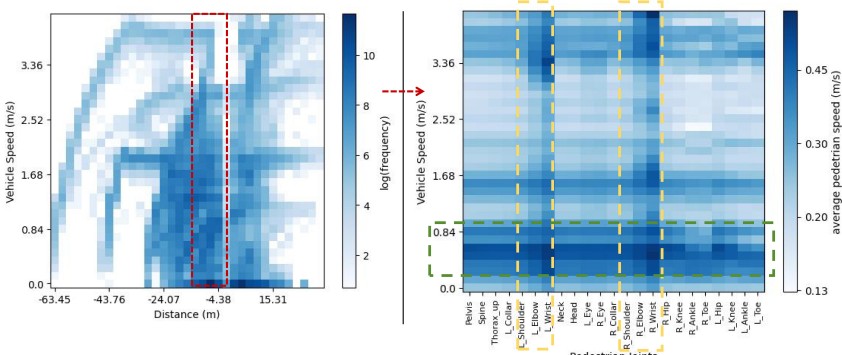

Figure 6: Left: the distribution of vehicle speeds and the distance between pedestrians and vehicles. The right box highlights scenarios when vehicles are approaching. Right: the relationship between vehicle speeds and pedestrian joint speeds when vehicles are approaching. The yellow box indicates that pedestrians tend to use their upper body to signal vehicles. The green box shows that pedestrians tend to move quickly when vehicles are moving slowly.

number of subjects, scenarios, sequences of third-person and egocentric perspective videos, and frame rates, are presented in Table 2. Comprehensive lists of the weather conditions and actors used, along with detailed illustrations of each designed scenario, are provided in the *supplementary materials*.

One of the distinctive features of our dataset is the real-time response of pedestrians under a variety of scenes, including rare and costly scenarios. We have designed diverse traffic scenes (*e.g.*, pedestrians may cross an intersection, seek assistance from passing vehicles, witness a traffic accident, *etc.*). Within each scene, multiple vehicles are introduced; some operate in autonomous modes to interact with pedestrians, while others follow predefined scripts according to various modes we established. We analyzed the distribution of vehicle speeds and the distances between pedestrians and vehicles in our dataset, as illustrated in Fig. 6, highlighting the diverse circumstances provided for pedestrian reactions. Additionally, we visualized the relationship between vehicle speed and pedestrian joint speed when the vehicle approaches the pedestrian, also shown in Fig. 6. From this analysis, several intriguing conclusions can be drawn: (a) Pedestrians tend to move fast when the vehicle is moving at low velocities while moving slowly when the vehicle is moving fast, indicating that people attempt to avoid potential danger from fast-moving vehicles or ignore slow-moving vehicles by moving quickly. (b) Pedestrians often use their upper limbs to signal vehicles, suggesting that body language is employed to communicate with drivers and avoid danger. We also analyzed the interaction processes of two examples, as shown in Fig. 7, and provided additional dataset statistics and insights in the *supplementary materials*. Furthermore, to explore human intentions, we employed the multimodal large language model GPT-4 to generate intention descriptions from third-person and ego-view images. These examples are likewise included in the *supplementary materials*.

For another distinctive features, our dataset consists of quantified data captured in extreme cases. We define the extreme cases as scenarios with an extremely low occurrence probability in real-world road conditions, where the cost of artificially constructing these scenes is often prohibitive. We provide more detailed discussions and examples of extreme cases in the *supplementary materials*.

## 4 EXPERIMENTS WITH DIFFERENT TASKS AND BENCHMARKS

Our dataset comprises various modalities, and accordingly, we divide it into three parallel subsets for validation. We evaluate these subsets using state-of-the-art baselines for 3D human body motion reconstruction tasks from third-person perspective, first-person perspective, and LiDAR perspective. We also study the real-synthesis domain gap with 3D pedestrian detection.

### 4.1 THIRD-PERSON PERSPECTIVE

**Baselines.** We randomly selected 20% of the sequences in the PMR dataset as our test set and evaluated a series of human mesh recovery methods. ROMP (Sun et al., 2021), HUMAN4D (Goel et al., 2023), HybrIK (Li et al., 2021), and PHALP+ (Rajasegaran et al., 2022) are designed to estimate human meshes from a single static camera and fail to recover global human trajectories, particularly

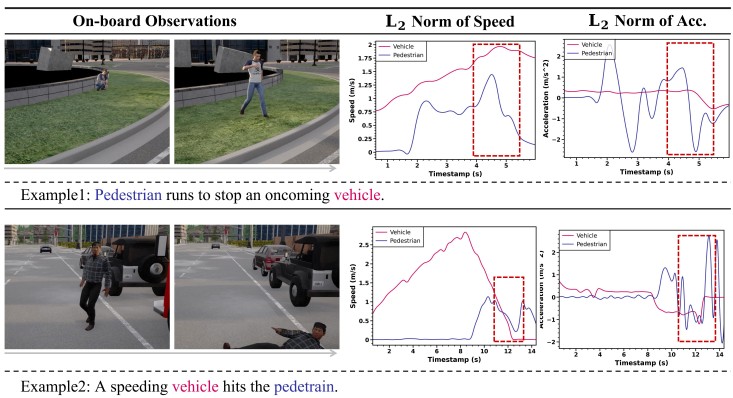

Figure 7: Pedestrian-vehicle interaction examples. In example 1, the pedestrian runs to stop an oncoming vehicle, and the vehicle slows down after seeing the pedestrian. In example 2, a collision occurs. Note that the left side provides the view from the vehicle (onboard observation).

Table 3: Comparison of human motion reconstruction baselines on third-person perspective RGB video and LiDAR modality on PMR dataset.

| Inputs | Methods | Space | Strategy | W-MPJPE↓ | WA-MPJPE↓ | Acc. Err.↓ | PA-MPJPE↓ |
|---|---|---|---|---|---|---|---|
| RGB | ROMP (Sun et al., 2021) | camera space | regression-based | 436.212 | 1963.060 | 1796.889 | 67.480 |
| | HUMAN4D (Goel et al., 2023) | | regression-based | 1048.893 | 289.565 | 304.739 | 63.108 |
| | HybrIK (Li et al., 2021) | | optimization-based | 2083.990 | 384.074 | 639.891 | 63.367 |
| | PHALP+ (Rajasegaran et al., 2022) | | optimization-based | 1156.317 | 301.723 | 495.168 | 62.463 |
| RGB | TRACE (Sun et al., 2023) | world space | regression-based | 1258.798 | 549.745 | 6095.813 | 66.718 |
| | GLAMR (Yuan et al., 2021a) | | optimization-based | 702.130 | 320.728 | 142.576 | 70.694 |
| | SLAHMR (Ye et al., 2023b) | | optimization-based | 378.358 | 183.924 | 151.024 | 73.518 |
| LiDAR+RGB | LPFormer* (Ye et al., 2023a) | world space | - | 71.254 | 60.377 | 44.905 | 57.067 |

Note: W-MPJPE, WA-MPJPE, and PA-MPJPE reported in $mm$, Acc. Err. reported in $mm/s^2$. *: modified LPFormer with RGB+Lidar inputs.

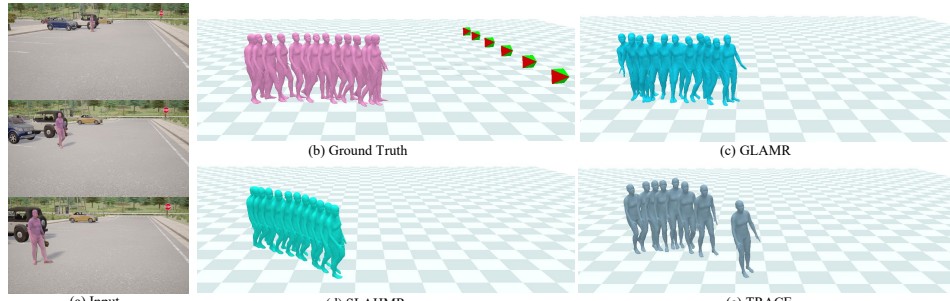

Figure 8: Qualitative evaluation of third-person perspective pedestrian reconstruction in global coordinate. (a) Input video with groundtruth SMPL-X. (b) Reconstruction groudtruth in global coordinate. (c)-(e) Results from baseline methods.

when the camera is moving. In contrast, GLAMR (Yuan et al., 2021a), SLAHMR (Ye et al., 2023b), and TRACE (Sun et al., 2023) account for camera motion, enabling the reconstruction of global human trajectories and more accurate estimation of global human motion in world coordinates. GLAMR employs a multi-step process to infer the overall human trajectory from local 3D poses, which are relative to the root and estimated frame-by-frame. SLAHMR utilizes a multi-stage optimization approach that integrates structure from motion with human motion priors to decouple human motion from camera motion, thereby computing 4D human trajectories in global coordinates. TRACE leverages scene information and 3D human motions using a 5D representation, exploiting all temporal cues to facilitate end-to-end training.

**Metrics.** For human pose estimation, following (Ye et al., 2023b), we adopt PA-MPJPE, Acceleration Error (Acc. Err.) (Kocabas et al., 2020; Yuan et al., 2021b), World PA Trajectory - MPJPE (WA-MPJPE) and World PA First - MPJPE (W-MPJPE). Compared with PA-MPJPE and Acc. Err., the two metrics commonly used on human pose estimation, WA-MPJPE and W-MPJPE pay more attention

Table 4: SLHAMR (Ye et al., 2023b) performance comparison under single and multiview.

| Camera Quantity | Moving modes | | Camera Relation | W-MPJPE↓ | WA-MPJPE↓ | Acc. Err.↓ | PA-MPJPE↓ |
|---|---|---|---|---|---|---|---|
| | Cam 1 | Cam 2 | | | | | |
| 1 | sta. | - | - | 267.416 | 168.427 | 158.734 | 83.499 |
| 2 | sta. | sta. | cooperative | **212.577** | **123.633** | **136.043** | **64.863** |
| 1 | mov. | - | - | **415.973** | **238.442** | **140.323** | 78.639 |
| 2 | mov. | mov. | cooperative | 463.104 | 239.368 | 144.944 | **77.199** |
| 1 | sta. | - | - | **267.416** | 168.427 | 158.734 | 83.499 |
| 2 | sta. | mov. | independent | 283.836 | **135.573** | **136.995** | **68.761** |
| 1 | mov. | - | - | **415.973** | 238.443 | 140.323 | **78.639** |
| 2 | mov. | sta. | independent | 509.428 | **225.854** | **136.955** | 79.951 |
| 1 | mov. | - | - | **416.834** | 220.676 | 129.950 | 80.238 |
| 2 | mov. | mov. | independent | 430.483 | **200.767** | **128.687** | **80.039** |

Figure 9: Qualitative comparison of SLAHMR in two scenarios: a) moving primary camera with static secondary camera, and b) static primary camera with moving secondary camera. Enforcing 2D joint projections in a single view fails to recover accurate 3D poses in both cases, while additional views help resolve the 2D-to-3D lifting issue.

to the global trajectories estimation, which are more adaptable to our tasks. To be more specific, WA-MPJPE computes MPJPE (Ionescu et al., 2014) after aligning the entire trajectories of the prediction and ground truth using Procrustes Alignment. W-MPJPE computes MPJPE with only the first frame of the prediction and ground truth aligned.

**Baselines Evaluation & Discussion.** The quantitative and qualitative results of human mesh recovery (HMR) methods on third-person perspective videos are presented in Table 3 and Fig. 8, respectively. It is evident that HMR methods operating in camera space yield higher PA-MPJPE results, yet significantly lower values in other metrics. This indicates that while these methods can achieve accurate pose estimation in local frames, they fail to reconstruct global trajectories in world coordinates. In contrast, HMR methods that account for world space demonstrate substantial improvements in overall performance, particularly in W-MPJPE and WA-MPJPE metrics. This underscores the importance and value of global human mesh recovery from moving cameras. However, TRACE exhibits suboptimal performance on our dataset, which we attribute to its distinct learning strategy. GLAMR and SLAHMR employ multi-stage optimization for each sequence, which, although time-consuming, enhances performance. Conversely, TRACE utilizes an end-to-end pipeline that is significantly faster but heavily dependent on the quantity and attributes of the dataset. The scarcity of comprehensive datasets in this domain contributes to the limited success and poor performance of regression-based global HMR methods. Consequently, our dataset offers considerable potential and opportunities for the development of end-to-end global HMR models. **Multi-View Evaluation.** The PMR dataset is the first to provide synchronized multi-view videos captured from sensors with varying moving modes and perspectives. This offers an opportunity to explore human motion reconstruction in more generic situations where both the number of cameras and camera movements are arbitrary. Intuitively, multi-view videos can enhance the accuracy of human pose and global trajectory reconstruction by providing additional prior knowledge and constraints, thereby mitigating the challenges associated with dimensionality lifting. Fully leveraging multiple RGB videos, regardless of their capturing modes, becomes a crucial issue. To address this, we established a baseline by adapting the SLAHMR method to accommodate multiple RGB video inputs. Detailed methodology and modifications are provided in the *supplementary materials*.

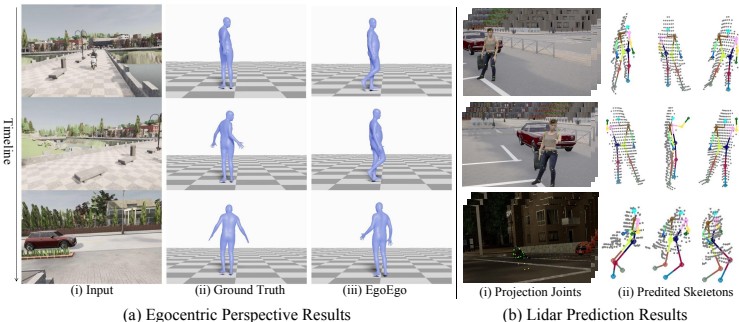

(i) Input      (ii) Ground Truth      (iii) EgoEgo        (i) Projection Joints     (ii) Predited Skeletons

(a) Egocentric Perspective Results            (b) Lidar Prediction Results

Figure 10: (a) Visualisation of egocentric perspective human mesh recovery (EgoEgo) results on our dataset. (b) LiDAR prediction results. (i) Ground truth (green) and predicted (yellow) projections of human joint keypoints onto the image. (ii) Original prediction of keypoints and skeleton, with gray points for the input LiDAR.

We take double-view as an example and categorize it into five groups based on whether the two cameras are static or moving, and whether they are independent or cooperative (stereo). For each group, we randomly select 40 sequences and perform two tasks: (i) the original SLAHMR, which only uses videos captured by the primary camera (cam 1) as input, (ii) our adapted SLAHMR, which takes videos from both the primary camera (cam 1) and the secondary camera (cam 2) as input. The results are presented in Table 4 and Fig. 9. Our results indicate that most cases benefit from the additional videos. However, when the primary camera is moving, the results of the multi-view SLAHMR sometimes are similar to or even worse than the original SLAHMR. This is because our multi-view pipeline relies on the global human trajectory recovery results in the initial optimization stage to align the coordinates of the two camera motion estimations. When the primary camera is moving, human trajectory results are more challenging to estimate accurately, leading to a series of errors in coordinate transformation and subsequent optimization stages, which are primarily based on the 2D keypoints projection loss from both views. The interference caused by the secondary camera can outweigh the positive effects, adversely affecting the final results. Therefore, maximizing the useful information from additional cameras while minimizing interference remains a significant challenge. Our PMR dataset provides an ideal foundation for investigating this issue.

## 4.2 EGOCENTRIC PERSPECTIVE

The egocentric perspective, which reveals pedestrian behavioral intentions, is one of the key advantages of using the mixed-reality platform for data collection. We evaluate the motion estimation from egocentric video on our PMR dataset using EgoEgo (Li et al., 2023a), the SOTA method to our knowledge. EgoEgo leverages SLAM (Teed & Deng, 2021) and transformer-based models (Vaswani et al., 2017) to estimate head motion from egocentric video through HeadNet and Gravity Net, and then generates full-body motion using a diffusion model (Tevet et al., 2022) conditioned on the predicted head pose. We trained the HeadNet on our dataset and employed the Gravity Net and diffusion model pretrained on AMASS, following the evaluation steps outlined in (Li et al., 2023a) or other datasets. We report comparisons between our dataset and existing datasets (*i.e.*, ARES (Li et al., 2023a), Kinpoly-MoCap (Luo et al., 2021), and GIMO (Zheng et al., 2022)) in Table 5. The evaluation metrics include Head Orientation Error ($\mathbf{O}_{head}$), Head Translation Error ($\mathbf{T}_{head}$), MPJPE, and Acc. Err., providing a comprehensive assessment of both head pose prediction and full-body motion estimation. As shown in Table 5, EgoEgo achieves acceptable results on our dataset, verifying the capability of our dataset to recover full-body human motions from egocentric VR videos. However, the performance of EgoEgo on PMR is slightly lower than on other datasets, and the reasons can be summarized as follows. First, the inherent uncertainty of the task shows we can infer reasonable estimates but cannot obtain precise ground truth, as illustrated in Fig. 10. Thus, the four metrics should be viewed as references. Additionally, our egocentric videos are captured using VR glasses in virtual environments, while most other datasets use head-mounted cameras in real-world settings, potentially introducing a domain gap that affects model adaptability. Despite these challenges, we believe that recovering human motion from virtual egocentric videos is valuable for sim-to-real transfer learning, character design, and building the metaverse in simulators. To our knowledge, the PMR dataset is the largest to feature real-time captured egocentric videos and full-body human motion in SMPL-X format, significantly contributing to advancements in human motion reconstruction from such videos.

Table 5: Full-body motion estimation from egocentric video by EgoEgo on datasets.

| Dataset | $O_{head}\downarrow$ | $T_{head}\downarrow$ | MPJPE↓ | Acc. Err.↓ |
|---|---|---|---|---|
| ARES (Li et al., 2023a) | 0.20 | 148.0 | 121.1 | 6.2 |
| Kinpoly-MoCap (Luo et al., 2021) | 0.58 | 505.1 | 125.9 | 8.0 |
| GIMO (Zheng et al., 2022) | 0.67 | 356.8 | 152.1 | 10.4 |
| PMR (Ours) | 0.24 | 587.5 | 192.3 | 21.6 |

Table 6: cross-dataset evaluation on BEVStereo using different training data.

| Training data | Test on nuScenes (Caesar et al., 2020) | | | |
|---|---|---|---|---|
| | AP↑ | ATE↓ | ASE↓ | AOE↓ |
| 100% nuScens | 0.088 | 0.919 | 0.318 | 1.100 |
| 60% nuScens + 40% PMR | **0.094** | **0.875** | 0.318 | **1.088** |

## 4.3 LiDAR

In addition to RGB images, LiDAR point clouds are a widely used modality in pedestrian motion reconstruction, offering greater range and suitability for night scenes. While some datasets include 3D human pose annotations on LiDAR (Sun et al., 2019), acquiring paired LiDAR and RGB data simultaneously is challenging due to the different locations and alignments of point clouds and image textures. Furthermore, obtaining accurate 3D human pose annotations in such datasets is unlikely, often resulting in long-tail scenarios. To address this gap, the proposed PMR dataset includes high-quality RGB-LiDAR paired pedestrian motion annotations, with accurate 3D human pose annotations directly obtained from the MoCap system. To demonstrate the effectiveness of the LiDAR modality in the PMR dataset, we modified LPformer (Ye et al., 2023a), a leading solution for human LiDAR pose estimation, to incorporate both RGB and LiDAR inputs. We compared this modified approach with other state-of-the-art solutions that use only RGB input on the PMR dataset. As shown in Table 3 , the inclusion of the LiDAR modality enhances performance in 3D human pose estimation. However, certain long-tail scenarios, such as arm raising (second row in Fig. 10.b), sitting on the ground (third row in Fig. 10.b) cannot be estimated correctly.

## 4.4 Domain Gap Evaluation

While the mixed-reality data with high-quality, labor-free labels is a strength, the domain gap between simulated data and real-world data can be a concern. To demonstrate that our mixed-reality data can effectively serve the same purpose as real-world data in benefiting downstream tasks, we selected the stereo 3D detection task BEVStereo (Li et al., 2023b)as an example and conducted comparative experiments with the nuScenes dataset(Caesar et al., 2020). For this experiment, we focused solely on pedestrian detection, using two cameras and two sweep images. We first used 10000 images from nuScenes to train the model, then replaced 40% of the nuScenes data with our PMR data, and tested on 2000 samples from the nuScenes dataset. We evaluated the results using Average Precision (AP), mean Average Translation Error (ATE), mean Average Scale Error (ASE), mean Average Orientation Error (AOE) metic. As shown in Table 6, incorporating PMR data into the training process improved 3D pedestrian detection performance across all four metrics. This improvement may be due to the rare scenarios with high-quality annotations present in PMR, which help compensate for real-world cases, making 3D pedestrian detection more effective in the diverse nuScenes dataset. These results validate that the mixed-reality data from PMR can enhance downstream tasks, despite the domain gap between real and synthetic data. Additionally, we performed another cross-dataset evaluation on human motion reconstruction (HMR (Kanazawa et al., 2018)), a well-acknowledged benchmark in human motion analysis. Results for this evaluation are provided in the *supplementary materials*.

## 5 Conclusion

In this paper, we address the challenge of intention-aware pedestrian motion reconstruction from dynamic sensors, which is currently underexplored due to the lack of annotated data. We therefore introduce the PMR dataset, developed from a mixed reality platform. This dataset captures the intricate dynamics of pedestrian interactions with objects and vehicles, using the pedestrian's egocentric perspective and LiDAR modality for enhanced modeling accuracy. The high volume of data provides a rich foundation for modeling pedestrian intent through multi-view and multi-modal insights. We also conduct comprehensive benchmark assessments across third-person and egocentric perspectives, as well as RGB and LiDAR modalities, for the pedestrian motion reconstruction task.

ACKNOWLEDGEMENTS

We appreciate all the anonymous reviewers for their constructive suggestions on polishing this paper. The work was supported by the Shanghai Municipal Science and Technology Major Project (Grant No. 2021SHZDZX0102), and the National Natural Science Foundation of China (Grant No. 62302295) in part.

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
