# OpenReview forum: "Pedestrian Motion Reconstruction: A Large-scale Benchmark via Mixed Reality Rendering with Multiple Perspectives and Modalities"
_ICLR.cc/2025/Conference — ICLR 2025 Poster_

### Official Review · Reviewer_ZsYG · 2024-10-31

**Soundness:** 4
**Presentation:** 4
**Contribution:** 4
**Rating:** 6
**Confidence:** 5

**Summary:**

The paper introduces the Pedestrian Motion Reconstruction (PMR) dataset, a large-scale benchmark designed to advance autonomous driving safety by focusing on pedestrian intention and behavior. The dataset is created using a mixed reality platform that combines real-world realism with simulation accuracy, capturing pedestrian interactions with objects and vehicles from multiple perspectives and modalities. The PMR dataset includes data from 54 subjects across 13 urban settings with 7 objects, encompassing 12,138 sequences under diverse weather conditions and vehicle speeds. The paper also presents comprehensive benchmark assessments across different modalities to evaluate pedestrian motion reconstruction methods.

**Strengths:**

1. The PMR dataset is the first of its kind to incorporate multi-perspectives and multi-views for modeling pedestrian intention in diverse outdoor scenes, including rare scenarios like collisions with safety concerns.
2. The dataset provides a rich foundation for modeling pedestrian intent through insights from third-person perspective RGB videos, LiDAR data, and egocentric perspectives.
3. The mixed reality platform reduces data collection costs and risks while ensuring ground-truth alignment with the global coordinate system, capturing pedestrian interactions realistically.
4. The paper conducts thorough evaluations across different modalities, providing a valuable resource for assessing pedestrian motion reconstruction methods.
5. The dataset is publicly available, promoting further research and development in the field.

**Weaknesses:**

1. While the mixed-reality data offers high-quality labels, there may be a domain gap between simulated and real-world data, which could affect the generalizability of models trained on the PMR dataset. The domain gap not only lies the rendering but also the motion diversity.
2. The reliance on a mixed reality platform with VR headsets, MoCap systems, and the CARLA simulator might limit the reproducibility of the data collection process in other research settings.
3. The dataset is collected from a limited number of subjects, which may not fully represent the global diversity in pedestrian behavior and motion.

**Questions:**

1. How does the PMR dataset address the domain gap between simulated and real-world data, and what measures are taken to ensure the dataset's applicability to real-world scenarios? The author should discuss more about how could this dataset working together with real captured dataset, for example waymo-open-dataset.
2. What are the limitations of the current data collection process, and how might these be overcome in future work to increase the diversity and representativeness of the dataset?

---

> ### Author Response · Authors · 2024-11-25
>
> **P1 While the mixed-reality data offers high-quality labels, there may be a domain gap between simulated and real-world data, which could affect the generalizability of models trained on the PMR dataset. The domain gap not only lies the rendering but also the motion diversity.**
>
> We appreciate the reviewer’s concern about the domain gap between simulated and real-world data, particularly regarding motion diversity. While simulated data may exhibit limited variability due to constraints like indoor spaces, the mixed-reality data collection process introduced in our paper can address this limitation by integrating real-world human pose data and retargeting it to the avatar in the simulator. This process allows us to replay real-world motion sequences within the simulation, increasing the variability of the dataset and the generalizability of models trained with dataset collected from the mix reality platform.
>
> **P2 The reliance on a mixed reality platform with VR headsets, MoCap systems, and the CARLA simulator might limit the reproducibility of the data collection process in other research settings.**
>
> We consider our data capture system as the most suitable solution to overcome the limitations of real-world auto-driving dataset. The capturing platform is also one of the main contributions. It is the whole system that make our work unique to others. Therefore, the complication and reproducing difficulty is kind of inevitable. On one hand, we will keep finding simpler solutions under the same idea for easier reproducibility。On the other hand, we will open-source the data capture platform on our website and repository as much as possible, including source codes for scene designing, data capturing, data export, data synchronizing, data visualization, data utilization, as well as detailed instructional documents about how to set and utilize the platform. We will also keep active in our repository to assist for related platforms setup to ensure the reproduction of our work as well as advancement of related research.
>
> **P3 The dataset is collected from a limited number of subjects, which may not fully represent the global diversity in pedestrian behavior and motion.**
>
> We will keep maintaining and expanding the dataset with more subjects, as well as more scenarios and objects for subjects to interact with to improve the global diversity.

---

> ### Author Response · Authors · 2024-11-25
>
> **Q1 How does the PMR dataset address the domain gap between simulated and real-world data, and what measures are taken to ensure the dataset's applicability to real-world scenarios? The author should discuss more about how could this dataset working together with real captured dataset, for example waymo-open-dataset.**
>
> To validate the domain gap caused by mix reality data collection process, we conducted a cross-dataset validation experiment, as detailed in Table 2 of the supplementary materials. Specifically, we trained a Human Motion Reconstruction (HMR) method using 300,000 frames from the real-world Human3.6M dataset. We then created a hybrid training set by replacing 40% of the real-world data with frames from the PMR dataset and trained a new model from scratch. Results showed that this new model outperformed the original model trained only with Human3.6M dataset when tested on unseen real-world datasets (3DPW and MPI-INF-3DHP). This improvement demonstrates the diversity and complementary value that PMR brings to real-world scenarios.
>
> To work collaboratively with real-world datasets such as the Waymo Open Dataset, PMR can serve as a high-quality complement, particularly in rare and high-risk scenarios that are difficult or costly to capture in real-world settings. For example, PMR includes detailed annotations for complex interactions, such as pedestrian behaviors in extreme weather conditions or during near-collision events, which are underrepresented in real-world datasets like Waymo.
> By integrating PMR into training pipelines alongside real-world datasets, researchers can leverage the rich diversity and precise annotations of PMR to enhance model robustness and generalizability. For instance, using PMR data to pre-train models can help improve performance when fine-tuning on real-world datasets.
> This integration between PMR and real-world datasets can ultimately result in better performance in downstream tasks such as pedestrian intent prediction, multi-modal behavior modeling, and human-vehicle interaction understanding, even in rare and challenging scenarios.
>
> **Q2 What are the limitations of the current data collection process, and how might these be overcome in future work to increase the diversity and representativeness of the dataset?**
>
> The current data collection process is conducted within a constrained 6m x 8m indoor space. This setup inherently limits the duration of pedestrian-vehicle interaction time to approximately 10 seconds. As a result, the captured scenarios are relatively short and primarily focus on critical interactions, such as pedestrians 'appearing from a blind spot' or 'crossing an intersection'. While these scenarios are carefully designed to capture meaningful dynamics, the spatial and temporal restrictions reduce the opportunity to observe more naturalistic and prolonged behaviors.
>
> To address these limitations and enhance the dataset’s diversity and representativeness, we propose the following steps for future work: (1) We plan to move the data collection platform to a larger outdoor urban environment. This will enable the capture of longer and more complex pedestrian behaviors, providing a more comprehensive view of real-world interactions. (2) We aim to open source the data collection platform, inviting contributions from other research teams. Collaborators could add real-world objects, such as various types of vehicles, urban elements, and diverse environmental setups, further enriching the dataset’s diversity and applicability.

---

### Official Review · Reviewer_XgLT · 2024-11-02

**Soundness:** 3
**Presentation:** 3
**Contribution:** 3
**Rating:** 8
**Confidence:** 4

**Summary:**

This paper presents a large-scale pedestrian motion reconstruction (PMR) dataset developed using a mixed reality platform. It simulates the authenticity of the real world while reducing the costs and risks associated with dataset creation. The dataset includes third-person perspective RGB videos of moving vehicles, LiDAR data from the vehicles, and self-centered views of pedestrians, along with both unimodal and multimodal pedestrian pose estimation. This dataset contributes to the advancement of the community.

**Strengths:**

1.	The dataset is designed based on the simulation of pedestrian intentions in the real world and features a pipeline for pedestrian motion capture utilizing VR and MoCap technologies. It replicates various pedestrian actions found in real-world scenarios, providing a reference for dataset collection and production.
2.	The substantial amount of data offers a rich foundation for modeling pedestrian intentions through multi-view and multimodal insights.
3.	The dataset also simulates the interaction between pedestrians and vehicles under different environmental conditions, which is very suitable for practical applications.

**Weaknesses:**

1.	I noticed that interactions between pedestrians and objects are quite limited, with a greater focus on pedestrian-vehicle interactions. Did the authors consider including specific scenarios, such as pedestrian conversations, telephone calls, and the use of umbrellas on rainy days?
2.	The comparison datasets in the paper are not up-to-date. To strengthen the comparison with datasets gathered from real-world scenarios, you may refer to some of the most recent multimodal human body reconstruction datasets. such as RELI11D(CVPR2024) [1], HiSC4D (TPAMI 2024) [2].
3.	In the collection pattern of this dataset, the encounters between pedestrians and vehicles in the environment are all predetermined by fixed program designs, which may result in slightly limited comprehensiveness of the dataset. Have the authors considered incorporating randomness in procedural generation techniques to accommodate the diversity of interactions?


**Reference**

[1] RELI11D: A Comprehensive Multimodal Human Motion Dataset and Method, CVPR 2024

[2] HiSC4D: Human-centered interaction and 4D Scene Capture in Large-scale Space Using Wearable IMUs and LiDAR, PAMI 2024

**Questions:**

1. There are already numerous existing human motion datasets; could you provide the latest datasets for comparison?
2. The picture annotation text in Figure 6 is blocked by the picture.
3. What is the frequency of camera and Lidar acquisition?
4. The ground truth used in this article is sampled from the mocap system to 120hz, please explain the rationality of the GT.
5. What are the shape distribution and the speed distribution of virtual avatars?

---

> ### Author Response · Authors · 2024-11-25
>
> **P1 I noticed that interactions between pedestrians and objects are quite limited, with a greater focus on pedestrian-vehicle interactions. Did the authors consider including specific scenarios, such as pedestrian conversations, telephone calls, and the use of umbrellas on rainy days?**
>
> Thank you for the comment. In this paper, as an initial attempt, we included 12 rigid common objects (e.g., suitcases) for pedestrians to interact with. More complex behaviors, such as conversations, telephone calls, or using umbrellas on rainy days, are challenging to capture within the constraints of a limited indoor space. In future work, we plan to extend our dataset collection to outdoor scenes, allowing for more diverse and realistic pedestrian behaviors. Additionally, we aim to integrate our dataset with real-world pose datasets to capture a broader range of interactions, including more nuanced pedestrian-object and pedestrian-vehicle scenarios.
>
> **P2 The comparison datasets in the paper are not up-to-date. To strengthen the comparison with datasets gathered from real-world scenarios, you may refer to some of the most recent multimodal human body reconstruction datasets. such as RELI11D(CVPR2024) [1], HiSC4D (TPAMI 2024) [2].**
>
> Thanks for your advice. **We have updated latest relevant multi-modal human body reconstruction datasets in Table 1 in the submitted draft.** To name a few, RELI11D [1] involves four types of sensors: RGB cameras, LiDAR, Event cameras, and IMU measurements, facilitating human pose reconstruction with rapid, coherent, and complex movements that require precise location. However, compared with our PMR dataset, it still lacks synchronized third-view RGB videos captured from multiple moving cameras and egoview RGB video to explore more about the interacting modes between subjects and scenarios. By contrast, HiSC4D [2] provides diverse human motions, rich human-human interactions, and human-environment interactions, promoting future research of egocentric human interaction in large scenes. Nevertheless, our PMR dataset has a much larger scale, and put our more attention on auto-driving scenarios. What's more, HiSC4D only provides 3D scenes without any direct RGB videos, leading to inconvenient usage for nowadays research on human motion reconstruction from RGB sequences. Besides, EHPT-XC [3] aims to solve the human pose estimation under extreme conditions. Unlike the rare cases in our PMR dataset, extreme conditions in EHPT-XC mainly includes cases of motion blur and low-light, where the difficulty of pose estimation is tackled using additional event cameras. MMVP [4] enriches their multi-modal human motion dataset with foot pressure (contact) for better human reconstruction performance. HmPEAR [5] provides imagery and LiDAR point cloud with 3D human pose as well as action annotations to bridge the gap between the fields of 3D human pose estimation and human action recognition, which our PMR dataset are also capable of as well.
>
> **P3 In the collection pattern of this dataset, the encounters between pedestrians and vehicles in the environment are all predetermined by fixed program designs, which may result in slightly limited comprehensiveness of the dataset. Have the authors considered incorporating randomness in procedural generation techniques to accommodate the diversity of interactions?**
>
> Actually, we have already incorporated randomness in our pre-designed scenes by spawning auto-driving cars of a random quantity (from 2 to 5) at random positions (closed to pedestrians). Those auto-driving cars make distinct decisions based on different situations, which add enormous randomness and complexity. Also, the appearance and category of all vehicles in the scenes are randomly selected. For instance, we represent some rare cases at https://anonymous.4open.science/r/PMRDataset-104B , where example (a) (b) (c) are based on the same scripts and scenarios. However, there is a significant difference in the speed and appearance of the vehicles, leading to completely different effects and pedestrian actions.
> We will highlight this point in the paper and will make sure to include it in the next version.
> Moreover, we appreciate your suggestion and will explore introducing additional randomness in future iterations of the dataset. For instance, we plan to randomize the spawn positions of pedestrians, the states of non-auto-driving vehicles, and the behaviors of other NPCs in the environment, etc.

---

> ### Author Response · Authors · 2024-11-25
>
> **Q1 There are already numerous existing human motion datasets; could you provide the latest datasets for comparison?**
>
> As mentioned in P2, **we have updated Table 1 in the related works in the submitted pdf**, which provides a more comprehensive introduction and comparison about relevant datasets.
>
> **Q2 The picture annotation text in Figure 6 is blocked by the picture.**
>
> Thank you for pointing out. We have corrected this issue in the updated draft to ensure that the annotations are clear and fully visible.
>
> **Q3 What is the frequency of camera and lidar acquisition?**
>
> Both camera and LiDAR data are captured at an average frequency of 6.72 Hz to ensure real-time performance of the system and synchronization of data across all modalities. While this frequency is slightly lower compared to some real-world datasets due to the high computational load of the simulator, it remains smooth and sufficient for human observation. Example videos demonstrating this can be found at https://anonymous.4open.science/r/PMRDataset-104B. Additionally, this frequency is adequate for most model training pipelines, which often involve downsampling video data to lower frequencies during preprocessing. Although the current stability of our PMR capturing system is acceptable, we recognize this as an area for improvement and will continue to refine it in future work.
>
> **Q4 The ground truth used in this article is sampled from the mocap system to 120hz, please explain the rationality of the GT**
>
> The ground truth (GT) data is sampled at 120 Hz from the motion capture (mocap) system to ensure high temporal precision and accurately capture detailed human motion. Given that the RGB data has an average frequency of 6.72 Hz, we aligned the datasets by selecting the mocap frame closest in time to each RGB frame. This alignment preserves the temporal correspondence between the GT and RGB data, ensuring that the GT accurately represents high-frequency motion details despite the lower capture frequency of the RGB data. This approach maintains the integrity of the motion data and ensures that the GT remains a reliable and precise reference for training and evaluation.
>
> **Q5 What are the shape distribution and the speed distribution of virtual avatars?**
>
> We visualize the shape distribution in terms of SMPL-X as well as the speed distribution of virtual avatars **in Figure 9 in the supplementary materials**.
>
> **References**
>
> [1] RELI11D: A Comprehensive Multimodal Human Motion Dataset and Method, CVPR 2024
>
> [2] HiSC4D: Human-centered interaction and 4D Scene Capture in Large-scale Space Using Wearable IMUs and LiDAR, PAMI 2024
>
> [3] A Benchmark Dataset for Event-Guided Human Pose Estimation and Tracking in Extreme Conditions, NeurIPS 2024
>
> [4] MMVP: A Multimodal MoCap Dataset with Vision and Pressure Sensors, CVPR 2024
>
> [5] HmPEAR: A Dataset for Human Pose Estimation and Action Recognition, ACM MM 2024

---

> ### Comment · Reviewer_XgLT · 2024-11-26
>
> Dear Authors,
>
> Thanks for your response. My concerns are addressed. I have increased the rating to Accept.
>
> Regards,
>
> Reviewer

---

### Official Review · Reviewer_7xAe · 2024-11-03

**Soundness:** 3
**Presentation:** 3
**Contribution:** 3
**Rating:** 8
**Confidence:** 3

**Summary:**

The paper introduces the Pedestrian Motion Reconstruction (PMR) dataset, a large-scale benchmark designed to improve the reconstruction of pedestrian motion and intention for autonomous driving applications. The dataset is generated using a mixed-reality platform that combines real-world realism with simulation-based data collection, reducing costs and safety risks. PMR features multi-modal and multi-perspective data, including third-person RGB videos, LiDAR data, and egocentric perspectives from 54 subjects across 13 urban settings. The dataset provides 12,000+ sequences under various weather conditions and vehicle speeds, capturing complex pedestrian-vehicle interactions, including rare scenarios like collisions. The main contributions include: 1) a mixed-reality platform of collecting real-world like pedestrian motion and corresponding sensor data 2) benchmark evaluations across different modalities such as third-person/first-person RGB and LiDAR for reconstructing pedestrian poses, and 3) insights into pedestrian behavior in dangerous and rare scenarios.

**Strengths:**

- The proposed mixed-realtiy platform combining real-world MoCap and VR simulation is interesting, and could provide a safe and efficient way to collect real-world pedestrian motion data and rare scenarios.
- It also provides a good benchmark for global human and camera motion estimation, with motion distribution closer to the real-world. Previous dataset often use existing MoCap motion data which often cannot reflect real-world human motion distributions.
- The dataset also includes real-world object interactions and tracking.

**Weaknesses:**

- The main tasks of the benchmark seem to focus on human pose estimation, which I think discard most of the scene context such as vehicles, weather, etc. It does not evaluate the main strengths of the dataset, which includes pedestrian vehicle interactions. I think having some pedistrian behavior generation/forecasting tasks would be beneficial since it will measure models’ ability to capture vehicle pedestrain interactions in rare or dangerous scenarios.
- To validate the domain gap between the proposed synthetic dataset and real-world data, the paper should evaluate on its main tasks, human pose estimation. For example, using the proposed dataset as additional training to improve the human pose estimation performance on existing benchmarks (3DPW, EMDB, RICH etc.). It is a bit weird that the paper evaluates the domain gap on a complete different task of 3D object detection.
- Finally, it would be nice if the paper can provide videos to showcase the rare cases and highlight the quality of the dataset.

**Questions:**

- How does the dataset decide which objects should be placed in real-world, and which should be placed in simulation?
- The proposed mixed-reality data capture platform is pretty neat. Any plans to open source it to expand the dataset further?
- What instructions are given to the subject to interact with the VR environment?
- Are there any audios such as traffic sounds provided to the subject to let them behavior more realistically?

Typo:
- l363: “taccount”

---

> ### Author Response · Authors · 2024-11-25
>
> **P1 The main tasks of the benchmark seem to focus on human pose estimation, which I think discard most of the scene context such as vehicles, weather, etc. It does not evaluate the main strengths of the dataset, which includes pedestrian vehicle interactions. I think having some pedestrian behavior generation/forecasting tasks would be beneficial since it will measure models' ability to capture vehicle pedestrian interactions in rare or dangerous scenarios.**
>
> As part of our future research, we plan to disentangle attribute-related and attribute-unrelated features from pedestrian behavior using a Semantics-Guided Neural Network as the feature extractor. Specifically, a disentangled bottleneck is employed to separate attribute-related and attribute-unrelated features, guided by an attribute classifier. Novel pose sequences are then generated by combining the specific attribute embeddings from one sequence with the embeddings representing the remaining attributes from another sequence. For example, if we have subject a's motion under scene 1 and subject b's motion under scene 2, we are curious about how subject 1 will react under scene 2. **Preliminary results is shown in Figure 11 in the supplementary**, demonstrate the feasibility of this approach. We believe this work has the potential to create more diverse and realistic pedestrian motion patterns.
>
> **P2 To validate the domain gap between the proposed synthetic dataset and real-world data, the paper should evaluate on its main tasks, human pose estimation. For example, using the proposed dataset as additional training to improve the human pose estimation performance on existing benchmarks (3DPW, EMDB, RICH etc.). It is a bit weird that the paper evaluates the domain gap on a complete different task of 3D object detection.**
>
> We have evaluated the domain gap on both a 3D pedestrian detection task and a human pose estimation task (specifically, human motion reconstruction using a well-recognized HMR method). As mentioned in Section 4.4, the results of the HMR evaluation are provided **in Table 2 of the supplementary materials**.
>
> To validate the domain gap in human motion reconstruction, we conducted a cross-dataset validation experiment, detailed in Table 2 of the supplementary materials. Specifically, we trained a Human Motion Reconstruction (HMR) method using 300,000 frames from the real-world Human3.6M dataset. We then created a hybrid training set by replacing 40% of the real-world data with frames from the PMR dataset and trained a new model from scratch. Results showed that this new model outperformed the original model trained only on Human3.6M when tested on unseen real-world datasets (3DPW and MPI-INF-3DHP). This improvement demonstrates the diversity and complementary value that PMR brings to training.
>
> In addition to human motion reconstruction, we also evaluated the domain gap using a 3D pedestrian detection task. This additional evaluation reflects the versatility of the PMR dataset. Its accurate labels, large scale, and multimodal nature make it a valuable resource for studying pedestrian behavior in various scenarios, which highlights PMR's potential to address domain gaps in different tasks and its utility in advancing research across multiple fields.
>
> **P3 It would be nice if the paper can provide videos to showcase the rare cases and highlight the quality of the dataset.**
>
> We have put some rare cases videos on our repository. Please checkout the videos on https://anonymous.4open.science/r/PMRDataset-104B .

---

> ### Author Response · Authors · 2024-11-25
>
> **Q1 How does the dataset decide which objects should be placed in real-world, and which should be placed in simulation?**
>
> The decision of whether objects are placed in the real world or rendered in simulation depends on their interaction dynamics and relevance to the scenario. Objects that interact directly with pedestrians, such as bags, suitcases, and umbrellas, are placed in the real world to accurately capture their 6DoF motion and ensure realistic, high-quality human-object interaction. Conversely, static objects commonly found in urban scenarios, such as buildings, pathways, roads, and traffic, are placed in the simulation to efficiently recreate complex environments without requiring physical setups.
>
> For objects placed in the real world, we simultaneously render their counterparts in the simulation to ensure consistency. This requires corresponding 3D models of the objects. Reflective markers are attached to the surface of these real-world objects to capture their transformations and rotations during interaction. These captured transformations are then applied in real time to their corresponding 3D models within the simulation, as detailed in Appendix A.2. This approach ensures seamless integration of real-world object dynamics into the simulated environment, enabling a cohesive and realistic interaction between real and virtual elements.
>
> **Q2 The proposed mixed-reality data capture platform is pretty neat. Any plans to open source it to expand the dataset further?**
>
> Yes, we plan to open-source the data capture platform on our website and repository in a phased manner. This will include source codes for scene design, data capturing, data export, data synchronization, data visualization, and data utilization. Alongside this, we will provide detailed instructional documents to guide users on how to set up and effectively utilize the platform. We will also keep active in our repository to assist for related platforms setup to ensure the reproduction of our work as well as advancement of related research.
>
> **Q3 What instructions are given to the subject to interact with the VR environment?**
>
> We give minimal instructions to the subjects to preserve the authenticity and diversity of their actions. The primary instruction is an audio cue that informs the subjects when to start moving. This audio cue is automatically played alongside specific scenes. The motivation for this instruction is to ensure proper coordination between the subjects and the moving vehicles that capture the third-person RGB videos. By synchronizing their movement, we ensure that the subjects remain at an appropriate position relative to the vehicles, allowing them to appear at a suitable size within the camera frame.
>
> **Q4 Are there any audios such as traffic sounds provided to the subject to let them behavior more realistically?**
>
> Yes, audio cues have been incorporated to enhance the realism of the scenes and elicit more natural responses from the subjects. CARLA provides convenient Python APIs for playing sounds within the simulation, which we leverage during the design of specific scenarios. For instance, a loud crash sound is triggered during severe collisions, while honking sounds are played when a vehicle approaches the subject at a relatively high speed. These audio effects help create an immersive and realistic environment, encouraging subjects to react more authentically to the simulated scenarios.

---

> > ### Comment · Reviewer_7xAe · 2024-11-26
> > **Thanks for the response**
> >
> > I’d like to thank the authors for providing a detailed rebuttal, which addressed most of my concerns. Thus, I have increased my score to accept.

---

### Official Review · Reviewer_MpM7 · 2024-11-04

**Soundness:** 3
**Presentation:** 3
**Contribution:** 3
**Rating:** 6
**Confidence:** 3

**Summary:**

The paper presents the Pedestrian Motion Reconstruction (PMR) dataset, a large-scale, mixed-reality dataset created to support research on pedestrian motion reconstruction, particularly focusing on pedestrian intent and behavior in urban environments. This dataset combines real-world elements with virtual simulations, allowing for extensive data capture with reduced safety and cost risks compared to real-world experiments.

**Strengths:**

The wide variety of urban setting, and the real data collected also with over 50 VR participants for first person perspective and realistic motions.  It also includes LIDAR. Totaling 12,138 sequences. The validation and the labeling of the dataset is good. All in all makes a very complete contribution.

**Weaknesses:**

I don't see major weaknesses. However, there could be domain Gaps from the Egocentric Data in PMR as it is collected through VR headsets in a simulated setting, which differs from real-world egocentric videos captured with physical head-mounted cameras.
I wonder how something like this could be done more similar to Ergo4D.
This domain discrepancy could limit the effectiveness of models trained on PMR for real-world applications, as the simulated egocentric data may not generalize well to real-world environments. This isn't a major flaw, but perhaps limits the impact of the dataset.

**Questions:**

I would like to see in the discussion how the data could also be used for causal implementations. Beyond just the current dataset. Many simulators recreate events by replicating configuration files. In fact a prior interesting work on a similar space but running on Airsim instead of CARLA, was looking into creating realistic behaviours based on motion capture and personality recreation.

Wang, Cheng Yao, et al. "CityLifeSim: A High-Fidelity Pedestrian and Vehicle Simulation with Complex Behaviors." 2022 IEEE 2nd International Conference on Intelligent Reality (ICIR). IEEE, 2022.

---

> ### Author Response · Authors · 2024-11-25
>
> **P1 There could be domain Gaps from the Egocentric Data in PMR as it is collected through VR headsets in a simulated setting, which differs from real-world egocentric videos captured with physical head-mounted cameras.**
>
> In this paper, our focus is on accurately capturing what pedestrians see when interacting with their environment. For real-world applications in autonomous driving, third-person videos remain the primary focus, as they are directly aligned with the requirements of autonomous driving systems, which prioritize the vehicle's perspective over egocentric views. However, simulated egocentric videos offer distinct advantages in terms of flexibility and scalability. They can be easily extended or integrated with other datasets, such as human pose datasets, and replayed in a controlled simulation environment to facilitate diverse applications and experiments.
>
> There is an unavoidable domain gap between real-world and simulated egocentric videos. Unlike head-mounted cameras used in Ego4D, which may introduce a misalignment between the recorded video and the pedestrian's actual field of view, our approach ensures a more precise representation by directly obtaining the videos rendered in VR glasses.
> Additionally, capturing exactly what pedestrians see in real-world scenarios raises significant safety concerns, especially in rare or high-risk situations.
>
> **Q1 I would like to see in the discussion how the data could also be used for causal implementations.**
>
> To explore human intentions, we utilized the multimodal large language model GPT-4o to generate intention descriptions based on both third-person and ego-view images. Examples of these human intention descriptions are provided in Figure 8 of the supplementary materials. This demonstrates the potential of our dataset for understanding human intent and offers a solid foundation for future causal analysis and implementations. For instance, these descriptions could be integrated into causal reasoning frameworks to model how specific pedestrian behaviors influence vehicle responses or how environmental factors shape human decision-making in traffic scenarios. By bridging intent interpretation and causal modeling, our dataset opens up new possibilities for analyzing and implementing causal relationships in human-vehicle interaction systems.
>
> **Q2 Beyond just the current dataset. Many simulators recreate events by replicating configuration files. In fact a prior interesting work on a similar space but running on Airsim instead of CARLA, was looking into creating realistic behaviours based on motion capture and personality recreation.**
>
> Our data collection process is highly flexible and can be extended to other simulators, such as AirSim [1]. To support community efforts, we will open-source the configuration files as prior works [2], enabling replication of the collected scenarios and facilitating contributions to this dataset. Additionally, by integrating these simulated scenarios with real human motion dataset, we aim to further enhance the diversity and representativeness of the PMR dataset. **We’ve updated this in the future work section in the supplementary.**
>
> **References**
>
> [1] Airsim: High-fidelity visual and physical simulation for autonomous vehicles. S. Shah, Debadeepta Dey, Chris Lovett, and Ashish Kapoor. In International Symposium on Field and Service Robotics, 2017
>
> [2] Citylifesim: A high-fidelity pedestrian and vehicle simulation with complex behaviors. Cheng Yao Wang, Oron Nir, Sai Vemprala, Ashish Kapoor, Eyal Ofek, Daniel McDuff, and MarGonzalez-Franco. In 2022 IEEE 2nd International Conference on Intelligent Reality (ICIR), 2022
>
> [3] Synthetic data generation framework, dataset, and efficient deep model for pedestrian intention prediction.Muhammad Naveed Riaz, Maciej Wielgosz, Abel García Romera, and Antonio M López. In 2023 IEEE 26th International Conference on Intelligent Transportation Systems (ITSC), 2023

---

### Author Response · Authors · 2024-11-25

We appreciate sincerely for all your precious reviews and positive ratings. We have updated both the pdf file and supplementary materials, where we marked the revised and newly added parts in red. Please check out our latest version.

---

### Meta-Review · Area_Chair_Dj2t · 2024-12-20

**Metareview:**

The paper presents the Pedestrian Motion Reconstruction dataset , a large-scale benchmark aimed at enhancing the reconstruction of pedestrian motion and intention for autonomous driving. Using a mixed-reality, it combines real-world realism with simulation to collect diverse multi-modal data, including RGB videos, LiDAR, and egocentric perspectives, capturing sequences across various urban scenarios and rare interactions. Key contributions include the innovative data collection platform, benchmark evaluations across modalities, dataset to get insights into pedestrian behavior in complex and hazardous situations.

**Additional Comments On Reviewer Discussion:**

Most of the criticism raised by the reviewers center around the domain gap, and limited interaction with exogenous variables such as weather, vehicle interaction etc.

Nonetheless all the reviewers are leaning positive and indicate that this is a good contribution.

---

### Decision · Program_Chairs · 2025-01-22

Accept (Poster)